# Orientation Estimation of Abdominal Ultrasound Images with Multi-Hypotheses Networks

**Timo Horstmann**                                       horstmann@imfusion.com
**Oliver Zettinig**                                        zettinig@imfusion.com
**Wolfgang Wein**                                            wein@imfusion.com
**Raphael Prevost**                                      prevost@imfusion.com
*ImFusion GmbH*
*Munich, Germany*

**Editors:** Under Review for MIDL 2022

## Abstract

Ultrasound imaging can provide valuable information to clinicians during interventions, in particular when fused with other modalities. Multi-modal image registration algorithms however require a somewhat accurate initialization, which is particularly difficult to estimate for ultrasound images as their orientation is arbitrary and their content ambiguous (limited field of view, artifacts, etc.). In this work, we not only train neural networks to predict the absolute orientation of ultrasound frames, but also to produce a confidence for each prediction. This allows us to select only the most confident frames in the clip. Our networks are trained to produce multiple hypotheses using a simple yet overlooked meta-loss that is specifically designed to capture the ambiguity of the input data. We show on several abdominal ultrasound datasets that multi-hypotheses networks provide better uncertainty estimates than Monte-Carlo dropout while being more efficient than network ensembling. Generic, easy to implement and able to quantify both data ambiguity and out-of-distribution samples, they represent a preferable alternative to traditional baselines for uncertainty estimation. On a clinical test our method produces estimates within 20° of the true orientation, which we can use to improve the accuracy of a subsequent registration algorithm down to less than 10°.

**Keywords:** ultrasound, abdominal, deep learning, orientation, multi-hypotheses, confidence, uncertainty.

## 1. Introduction

Multi-modal image registration consists in aligning images obtained from different medical imaging modalities. A common application is abdominal image-guided surgery, where a preoperative magnetic resonance (MR) or computed tomography (CT) image has to be matched to a set of images, acquired just before or during the intervention, showing the current state of the patient. Ultrasound (US) imaging is particularly suitable for the latter since it is safe, cheap, can be used flexibly in real-time, and even tracked so that 3D volumes can be generated. However, multi-modal registration involving ultrasound is extremely challenging due to a number of factors: first, the images are noisy and contain various types of artifacts. Furthermore, the tissue gets deformed during the acquisition, since a certain pressure on the tissue is necessary to obtain good images. More importantly, an ultrasound frame recording only shows a restricted field of view which is not enough to cover the whole liver and, unlike CT or MR, is acquired in a completely arbitrary orientation.

For those reasons, all algorithms for US registration typically need to be initialized close enough to the true solution, i.e. within the optimizer's capture range. In (Müller et al., 2014) for instance, the authors trained models to segment some specific structure of interest in both modalities, and then register them rigidly using a global search. This however only translates the problem to label maps and the global search is not guaranteed to succeed. Furthermore, it enforces a prior on structures that are supposed to be visible and relevant.

Instead, inspired by previous work on standard plane detection (Baumgartner et al., 2017), we are interested in trying to directly learn the orientation of 2D ultrasound images. It has indeed been proven that neural networks can learn orientations of 2D slices in the context of MR brain (Hou et al., 2018) or CT thorax scans (Hou et al., 2017). More recently, similar work has been applied to fetal ultrasound imaging in (Namburete et al., 2018) and (Yeung et al., 2021) with good results. However, liver acquisitions are more challenging than fetal brain scans due to the higher variability and the partial field of view.

For our application, a key observation is that every ultrasound *sweep* is composed of multiple 2D frames - each of those frames could provide its own estimate of the orientation. Yet some frames contain more relevant information than others: e.g. large artifacts can be present and hinder visibility, or the shape of the cross-section of the liver gland might be ambiguous. We therefore cannot expect all predictions to be equally accurate and would like to be able to automatically select and use only the least ambiguous ones.

Due to the growing interest in deep learning explainability, uncertainty estimation has been a very active field of research and provides models the ability to express their confidence on their own predictions. Despite the large number of methods (Gawlikowski et al., 2021), the two most popular approaches remain Monte-Carlo (MC) Dropout (Gal and Ghahramani, 2016) and network ensembling (Lakshminarayanan et al., 2017), probably due to their simplicity.

Recently, (Rupprecht et al., 2017) and then (Manhardt et al., 2019) introduced a simple and elegant approach with *multi-hypotheses networks* to address the problem of 6D pose estimation from RGB images, and in particular to deal with the inherent ambiguity of the dataset (for instance the symmetry of an object prevents to estimate a single pose). This method appears adapted to our problem, since the information from a single slice might not always be sufficient to regress the orientation. Surprisingly missing from the otherwise extensive survey in (Gawlikowski et al., 2021), this approach did not seem to receive a lot of attention, and has not really made its way into the biomedical field yet, apart from two applications: out-of-sample detection in histopathology (Linmans et al., 2020) and probabilistic segmentation (Kohl et al., 2018). Our goal is to investigate its potential for a different kind of application and compare it to the other baselines.

The main contributions of this paper are therefore to:

1. Propose a simple method to regress the global orientation of a series of tracked 2D ultrasound images and improve a subsequent multi-modal, intensity-based registration algorithm.

2. Investigate the advantages of multi-hypothesis networks over MC Dropout and ensembles on a regression task for ultrasound images.

## 2. Methods

### 2.1. Frame Orientation Prediction

We use unit quaternions to define the orientation of the ultrasound probe as they provide a simple and compact representation. In contrast to Euler angles, the quaternion representation does not suffer from Gimbal lock or discontinuities; they are also simpler to deal with than rotation matrices since normalizing them is easier than ensuring matrix orthogonality. The angular distance denoting the angle of rotation required to get from one quaternion to another is given by

$$d(\boldsymbol{q_1}, \boldsymbol{q_2}) = cos^{-1}(2\langle \boldsymbol{q_1}, \boldsymbol{q_2}\rangle^2 - 1). \tag{1}$$

A convolutional neural network like ResNet-18 (He et al., 2016) can be fine-tuned to minimize this distance between its 4-valued output (in order to satisfy the constraint of unit quaternions, we also apply a normalization at the end) and the ground truth quaternion.

As we mentioned in the introduction, there exist other ways to formulate this problem, e.g. (Hou et al., 2017, 2018). However, the multi-hypothesis approach that will be described in the next subsection is fully generic and could also be applied to such formulations.

### 2.2. Confidence Estimation via Multi-Hypothesis Networks

#### 2.2.1. MULTI-HYPOTHESIS NETWORK WITH META-LOSS

As introduced in (Manhardt et al., 2019), a neural network can be effortlessly changed into a multi-hypotheses network by replacing its last layer with several parallel heads. Each head consists of a fully connected layer, in our case with an output dimension of 4 (i.e. a quaternion). For one image $\boldsymbol{I}$, the network thus returns $M$ orientation hypotheses $\boldsymbol{f}(\boldsymbol{I}) = (\boldsymbol{f}_1(\boldsymbol{I}), ..., \boldsymbol{f}_M(\boldsymbol{I}))$. The key part of the method is the use of a meta-loss that considers all hypotheses:

$$\mathcal{L}(\boldsymbol{f}(\boldsymbol{I}), \boldsymbol{q}) = \left(1 - \varepsilon\frac{M}{M-1}\right)\min_{i=1,...,M} d(\boldsymbol{f}_i(\boldsymbol{I}), \boldsymbol{q}) + \frac{\varepsilon}{M-1}\sum_{i=1}^{M} d(\boldsymbol{f}_i(\boldsymbol{I}), \boldsymbol{q}) \tag{2}$$

where $d$ denotes the angular distance between quaternions (see Eq. 1) in our case (but could have been replaced by any other loss) and $\boldsymbol{q}$ is the corresponding ground truth quaternion.

In order to minimize this loss function, the network learns to spread out its answers for ambiguous images (so that at least one hypothesis is close to the ground truth, and therefore the first term has a low value), but collapses its outputs to one common answer for easier images (so that the second term is also minimized when possible). The hyperparameter $\varepsilon$ allows putting more emphasis on either term of the loss, but our experiments showed that its precise value is not critical.

The approach is simple to implement and, unlike methods such as Monte-Carlo dropout or ensembles, only requires a single forward pass to produce all hypotheses.

#### 2.2.2. UNCERTAINTY ESTIMATION

For each prediction, the network outputs $M$ hypotheses that can be reduced to a single result by computing their median but also, and more importantly, to compute a measure of uncertainty by quantifying the spread of those hypotheses.

In (Manhardt et al., 2019), the authors suggest to perform a singular value decomposition on the hypotheses. If the dominant singular values $(\sigma_i)_i$ are much greater than 0, the sample is considered to be ambiguous. To decide whether a prediction is certain enough, the authors utilized $\sigma_2$. Our experiments showed that other approaches (using $\sigma_1$ or computing the standard deviation) yield very similar results.

## 2.3. Initialization for Registration

So far we focused on the orientation of single frames. When we have a 3D tracked acquisition available, the relative 3D position/orientation of all frames with respect to each other are known. In that case, we wish to register the entire US acquisition, i.e. with all its frames, jointly and rigidly to a pre-operative volume. We describe here below how we compute a single rigid matrix, which could then be used as initialization for a registration algorithm.

**Rotation** Every frame of the sweep is fed into the neural network, and using the known relative orientations between the tracked frames, the entire sweep is aligned with respect to the prediction of the frame with lowest uncertainty.

**Translation** Once the rotation of the sweep is established, generating an estimate of the translation becomes easier. We train two additional segmentation U-Nets - one for ultrasound, one for the pre-op modality - to segment relevant anatomical structures that are visible in both images (diaphragm, gall bladder, vena cava); we then find the best translation that make those two prediction maps match. Such an approach was already applied in the literature (Müller et al., 2014), but here we only use it to estimate the translation. In the experiments section, we will compare our approach to directly estimating the global initialization from those prediction maps.

## 3. Datasets

Our experiments are based on three different datasets with increasing complexity.

**Dataset A - Toy dataset** The first dataset was synthetically generated and used to initially validate the method. 100 3D binary masks representing livers were selected from the LiTS public dataset (Bilic et al., 2019). For each binary mask, we extract 6000 2D frames by reslicing it from random directions with random offsets - the ground truth is thus the known orientation along which each frame has been sampled. While not realistic, this dataset is still related to our problem and contains inherent ambiguity without suffering from any artifacts; it is therefore suited for an initial validation.

**Dataset B - Experimental** For the purpose of this study we acquired a dataset of freehand ultrasound sweeps on 16 healthy volunteers. We used a research ultrasound system (Cephasonics Cicada) with a convex probe equipped with active targets tracked via an infrared system. During the acquisition, special care was taken to ensure a great variety in terms of the position and orientation of the probe when scanning the liver. The sweeps from all volunteers were acquired in the same supine position. For each volunteer, we acquired a reference sweep with a fixed start and end position;

those sweeps were used to register all volunteers with each other so that their coordinate systems match and are similar to the standard coordinate system of MR and CT images. In total, this dataset includes 28 545 tracked ultrasound frames.

**Dataset C - Clinical** The third dataset has been acquired on 16 patients before surgery with a different ultrasound system. No special protocol has been enforced during the acquisition, the operator simply acquired one or two sweeps to maximize tumor and liver gland visibility. This is the most challenging dataset since (i) images contain abnormal structures such as tumors, (ii) image quality is not optimal due to patient morphology and physical constraints, (iii) the number of data is smaller (247 frames per patient in average). The ground truth orientation has been obtained by manually registering every sweep to its pre-operative MR image.

Due to the relatively low number of volunteers/patients, all experiments on datasets B and C have been performed with 4-fold cross validation. Dataset A has been however randomly split into 78% training and validation. Every sweep is mirrored alongside the horizontal axis and appended to the original sweep, doubling the number of frames. Mirroring the frames simulates holding the probe the other way around (the ground truth labels for the mirrored frames are then adapted to take this orientation change into account).

## 4. Experiments and Discussion

### 4.1. Results on orientation estimation

#### 4.1.1. Comparison to baselines

Our networks predict $M = 10$ hypotheses and are trained with $\varepsilon = 0.6$. We compare them to several baselines: (1) a standard ResNet without any confidence estimation, (2) Monte-Carlo dropout models, in which we applied dropout on the penultimate layer during training and inference with a probability of 0.3, (3) network ensembles, for which we trained each network on a subset of the original dataset created by bagging. All networks are trained using the Adam optimizer with a learning rate of $10^{-4}$ and $\beta_1 = 0.9$ and $\beta_2 = 0.999$. Table 1 summarizes all metrics on the three datasets and illustrates the following observations:

- For all methods and all datasets, global errors (column 1) are significantly higher than errors from the most confident frames (column 2). This means that they are all able to select a better frame than the average via their confidence estimation.

- Angular errors are globally lower for Dataset C (clinical) than Dataset B (see column 1). Although surprising at first sight, this is most probably due to a reduced range of angles, whereas we purposely acquired frames with as much variability as possible in Dataset B.

- Multi-hypotheses networks provide uncertainties on par with ensemble models, as demonstrated by the mostly similar correlations (column 3). There is however a large difference with MC dropout whose uncertainties correlate much less.

- Since multi-hypotheses networks have several heads but a single backbone, they are naturally much more efficient than ensembles, for which the training and testing time

Table 1: Summary of evaluation metrics for all methods on the three different datasets. Angular errors are expressed in degree and averaged over the cases. The first column includes all frames of the sweep (so they represent the prediction error without taking the uncertainty estimates into account). The next column shows, for each case, the error of the frame with the lowest uncertainty and is therefore expected to be lower than the previous one. $\rho$ is the correlation between the uncertainty of the network prediction and the error of its prediction. Training times represent one forward-backward pass, testing times one forward pass.

| Method [Dataset A] | Angular Error (avg. all frames) | Angular Error (most confident) | Correlation $\rho$ | Train. Time | Test. Time |
|---|---|---|---|---|---|
| ResNet-18 | 31.68 | N/A | N/A | $25ms$ | $1.7ms$ |
| MC-Dropout (0.3, 10) | 32.37 | 15.01 | 0.224 | $25ms$ | $3.4ms$ |
| MC-Dropout (0.3, 20) | 32.14 | 12.89 | 0.227 | $25ms$ | $4.3ms$ |
| Ensemble (5) | 30.94 | 16.9 | 0.449 | $125ms$ | $10ms$ |
| Ensemble (10) | 29.22 | 11.0 | 0.483 | $250ms$ | $20ms$ |
| Multi-Hyp (10) | 26.07 | 11.23 | 0.428 | $35ms$ | $3.4ms$ |
| **[Dataset B]** | | | | | |
| ResNet-18 | $39.68 \pm 4.37$ | N/A | N/A | | |
| MC-Dropout (0.3, 10) | $39.94 \pm 4.29$ | $29.17 \pm 9.22$ | $0.224 \pm 0.05$ | | |
| MC-Dropout (0.3, 20) | $39.86 \pm 4.29$ | $22.23 \pm 4.01$ | $0.249 \pm 0.06$ | | |
| Ensemble (5) | $38.42 \pm 4.38$ | $21.52 \pm 2.89$ | $0.368 \pm 0.07$ | (see above) | |
| Ensemble (10) | $36.99 \pm 4.20$ | $23.14 \pm 3.94$ | $0.376 \pm 0.06$ | | |
| Multi-Hyp (10) | $39.24 \pm 5.03$ | $18.58 \pm 4.92$ | $0.430 \pm 0.04$ | | |
| **[Dataset C]** | | | | | |
| ResNet-18 | $20.38 \pm 1.69$ | N/A | N/A | | |
| MC-Dropout (0.3, 10) | $21.22 \pm 2.24$ | $14.12 \pm 1.78$ | $0.362 \pm 0.09$ | | |
| MC-Dropout (0.3, 20) | $21.09 \pm 2.18$ | $15.62 \pm 4.31$ | $0.394 \pm 0.11$ | | |
| Ensemble (5) | $25.46 \pm 2.16$ | $13.40 \pm 2.09$ | $0.564 \pm 0.13$ | (see above) | |
| Ensemble (10) | $24.79 \pm 1.75$ | $14.47 \pm 1.88$ | $0.511 \pm 0.13$ | | |
| Multi-Hyp (10) | $20.07 \pm 2.04$ | $14.50 \pm 2.29$ | $0.411 \pm 0.06$ | | |

increases linearly with the number of networks (column 4-5). Overall, they therefore seem to achieve the best trade-off between performance and simplicity.

### 4.1.2. ANALYSIS OF THE UNCERTAINTY ESTIMATION

We then extend our analysis on the uncertainty estimation. First we show in Figure 1 the relationship between the uncertainties estimated by the multi-hypotheses network and the error of their corresponding predictions. In plot (a), we distribute all frames of Dataset B into different bins based on their uncertainty estimate and we compute the average error of each bin. We observe a clear tendency: frames with a high uncertainty have indeed a much higher error over the whole range of values. Plot (b) shows a frame-by-frame comparison for a sweep where the correlation was the highest (0.8), with 3 extracted frames that have different levels of uncertainties. This particular sweep has been acquired with a focus on the right kidney. While the kidney is well contrasted and in a longitudinal view in the first sweep frames, the angle changes and the contrast with the liver gland decreases: the network then becomes less certain about its prediction.

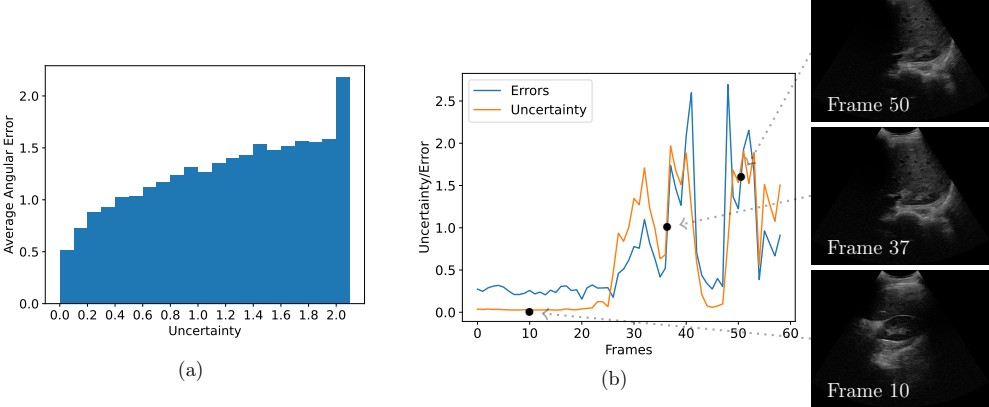

Figure 1: (a) Average errors for all frames of Dataset B as a function of the uncertainty as computed by the multi-hypotheses network. (b) Uncertainty vs. angular error for all frames of the sweep with maximum correlation.

Next, in order to understand more about the uncertainty that multi-hypotheses networks capture, we run on dataset B the networks trained on Dataset C (and vice versa). While the networks do not seem to generalize to a much different dataset, it is interesting to see that the average uncertainty score is significantly higher for out-of-distribution samples (see Table 2). This means that multi-hypotheses networks can at least detect images from a different source, even though they were not trained for this specific purpose.

Table 2: Average uncertainty for models trained on one dataset and applied to another.

| Model | Dataset B | Dataset C |
|---|---|---|
| Multi-Hyp trained on B | 0.237 | 0.453 |
| Multi-Hyp trained on C | 0.72 | 0.118 |

### 4.2. Usage for initialization of registration

Finally, we study on Dataset C the impact of our orientation estimation algorithm on a full pipeline of registration to a pre-op image. We use the method described in Section 2.3 to initialize an off-the-shelf multi-modal registration algorithm available in the ImFusion Suite (ImFusion GmbH, Munich, Germany) software, which automatically registers US and MR volumes with a deformation model on top of a rigid transformation by maximizing an LC2 similarity measure (Wein et al., 2013). This registration algorithm yields good results but requires a close initialization to converge.

As baseline, we use the result of the global search with the segmentation maps (i.e. without any pre-alignment with orientation estimation) as initialization of the registration. Errors on translation and rotation with respect to the ground truth are summarized in the boxplots of Figure 2. They demonstrate that adding our orientation estimation method at the beginning of the pipeline provides a better initialization (median error goes from 24.4 to 16.3 degrees), which in turn allows the registration algorithm to converge closer to the

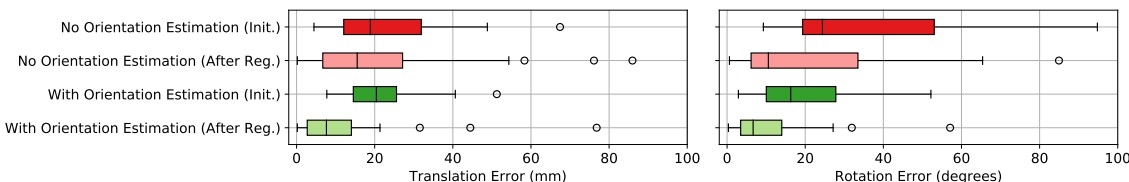

Figure 2: Comparison of the translation and rotation errors on Dataset C with and without our orientation estimation, and before versus after full image-based registration.

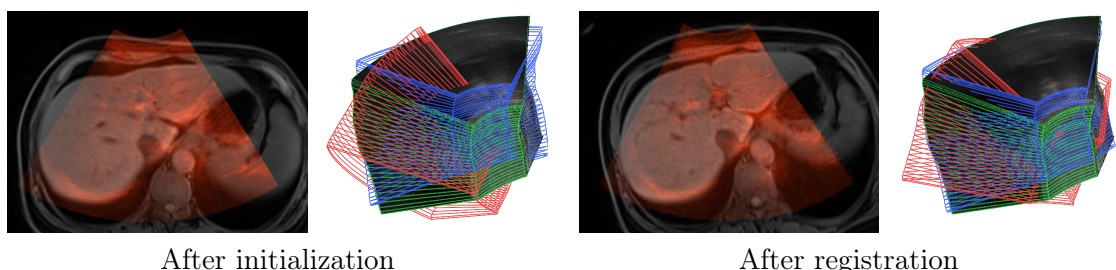

After initialization       After registration

Figure 3: Visualization of an US sweep after initialization (left) and after registration (right). The US image overlaid onto the MR scan corresponds to the proposed method. In the 3D view, red corresponds to the baseline (no orientation estimation), blue to our method (orientation estimation) and green to the ground truth.

ground truth. The final median errors are 7.6 mm and 6.7 degrees, with much fewer outliers than the baseline. Visual results are also shown in Figure 3.

## 5. Conclusion

In this paper, we addressed the problem of orientation regression for ultrasound images. In particular, we investigated the potential of multi-hypotheses networks for uncertainty estimation. While they do not significantly outperform network ensembling, we showed that they still produce useful estimates of the uncertainty at a much lower cost (at both training and testing time). Therefore, they probably represent a preferable baseline over MC dropout or network ensembling.

On the basis of this feasibility demonstration, numerous areas for improvement can be suggested. A landmark-based formulation of the problem as in (Hou et al., 2017) could for instance provide more accurate orientation predictions than quaternions. Moreover, adding temporal information during training and inference (e.g. using neighboring frames), would probably help generating smoother and more realistic results. Finally, additional work should be dedicated to improve the generalization of the network across datasets.

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

## Appendix A. Implementation of Multi-Hypotheses Networks

In order to emphasize the simplicity of implementation of multi-hypotheses networks and their associated meta-loss, we show here some exemplary Pytorch code. Note that the implementation is independent from the backbone network and the original loss functions. In our case, `hyp_dim` is 4 since each hypothesis is four-dimensional (a quaternion).

```python
class MetaLoss():
    def __init__(self, loss: nn.modules.loss._Loss, epsilon: float):
        self.loss, self.eps = loss, epsilon

    def compute(self, pred: torch.Tensor, gt: torch.Tensor) -> torch.Tensor:
        M = pred.shape[1]
        losses = torch.stack([self.loss(pred[:,i,:], gt) for i in range(M)], dim=1)
        meta_loss = (1-(self.eps*M)/(M-1))*torch.min(losses, dim=1)[0] \
            + self.eps/(M-1)*torch.sum(losses, dim=1)
        return torch.mean(meta_loss)

class MultiHypothesisNetwork(nn.Module):
    def __init__(self, model: nn.Module, num_hyp: int, hyp_dim: int = 4):
        super(MultiHypothesisNetwork, self).__init__()
        self.model, self.M, self.D = model, num_hyp
        self.model.fc = nn.Linear(num_feat, self.M * self.D)

    def forward(self, x: torch.Tensor) -> torch.Tensor:
        return self.model(x).view(-1, self.M, self.D)

def compute_uncertainty(hypotheses: torch.Tensor) -> float:
    pca = sklearn.decomposition.PCA(n_components=2)
    pca.fit_transform(hypotheses.detach().cpu().numpy())
    return pca.singular_values_[1]
```

## Appendix B. Sample frames from the datasets

In Figure 4, we plot some example images from the different datasets used in the paper.

**Dataset A** is composed of slices of the 3D binary masks representing livers. The shape of the liver is clearly visible and since the slicing is done from random arbitrary angles, a lot of shape variations are present. In comparison to the other datasets, these toy examples do not suffer from artifacts or poor quality while still containing ambiguity.

**Dataset B** contains abdominal US acquisitions, which are inherently noisy and suffer from artifacts. In each frame image, a part of the liver and blood vessels are visible. Additional structures present in some images include kidney, gall bladder, diaphragm. Such structures can provide additional clues for the network.

**Dataset C** Due to being obtained with a another ultrasound system, the frames have a different appearance (texture filtering, etc.). Since the acquisition happened in a clinical setting, a small range of angles was considered, making the content of the frames less variable as in dataset B.

Figure 5 shows the ground truth distribution of the Euler angles. Dataset A is by construction uniformly distributed over all possible angles. Dataset B has also been acquired

with a wide variety of angles but some orientations were simply not feasible due to e.g. occlusions of the ribs or patient position. Dataset C shows a more limited distribution due to its relatively small size and the fact that it was acquired independently for clinical purposes.

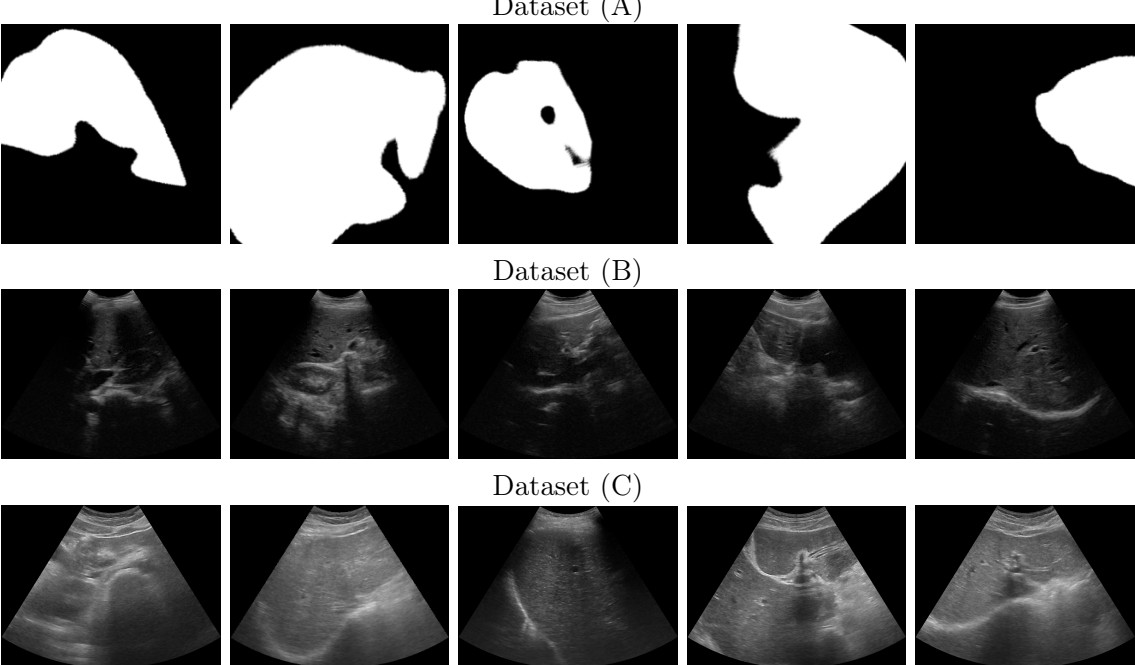

Figure 4: Sample images from the different datasets.

## Appendix C. Hyperparameter analysis

In order to train a multi-hypotheses network, two hyper-parameters (namely number of hypotheses M and the weight $\varepsilon$ of the loss function) have to be selected. In Figure 6 we study the effect of those hyperparameters on the prediction performance and find, that different setups only have a marginal effect on the average angular error. Evaluating the impact on the correlation of error and uncertainty (Figure 6), we observe only small differences in the Spearman's correlation. The correlations also remain comparable to other baselines (see Table 1). In summary, multi-hypotheses networks seem relatively robust to their hyperparameters.

## Appendix D. Analysis on best acquisition angles

In this appendix, we aim at further analyzing the uncertainty of the network across the whole dataset, in order to get more insight about the visual cues used by the network. In particular, we investigate whether some orientations are overall easier to predict than others (such observations could help defining standard planes or setting guidelines for future acquisitions). To that end, we represent all possible orientations on a sphere (each section of

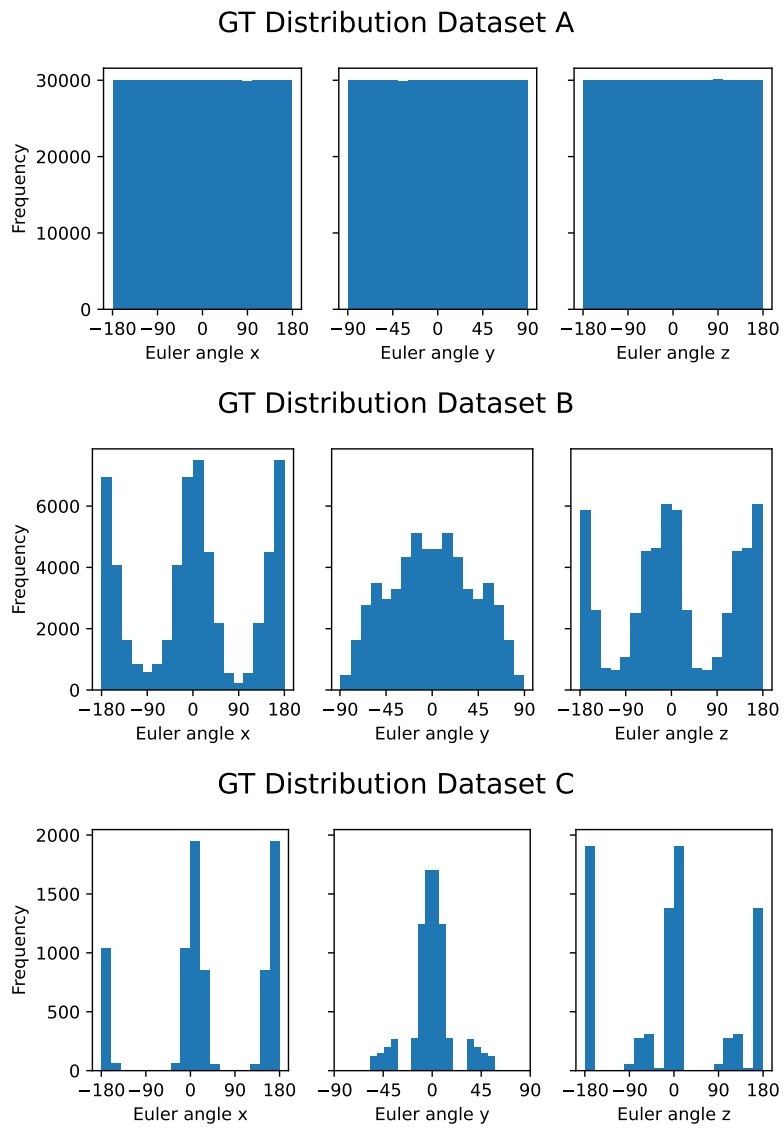

Figure 5: Histogram of Euler angles for the three datasets.

the sphere represents a bin around one particular orientation) and paint each section with the average error or the average uncertainty. We found and show in Figure 7 that there is a specific region for which both the errors and the uncertainty are low. This means that this range of angles should be preferred (or at least covered) during the image acquisition. Figure 7 also reports a representative subset of those images. Once again the kidney seems to play an important role, as we can see it in many images, in conjunction with the liver gland. We also notice the presence of large blood vessels which could also help the network.

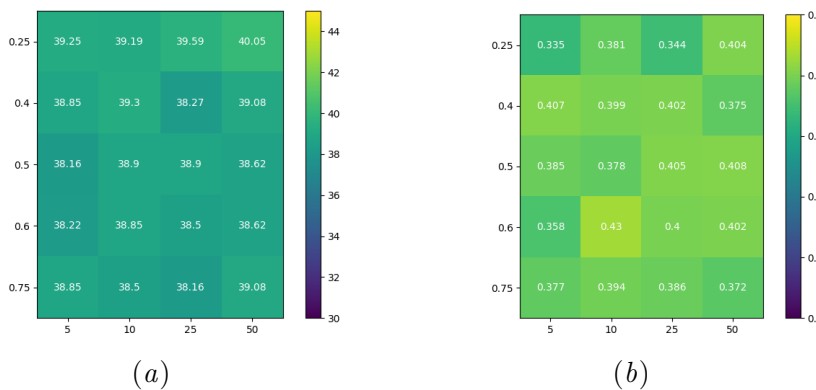

Figure 6: Robustness of method w.r.t hyperparameters M and $\varepsilon$. a) shows the robustness w.r.t. to the average error (in degree) over validation set, b) shows the Spearman's correlation for different hyperparameters.

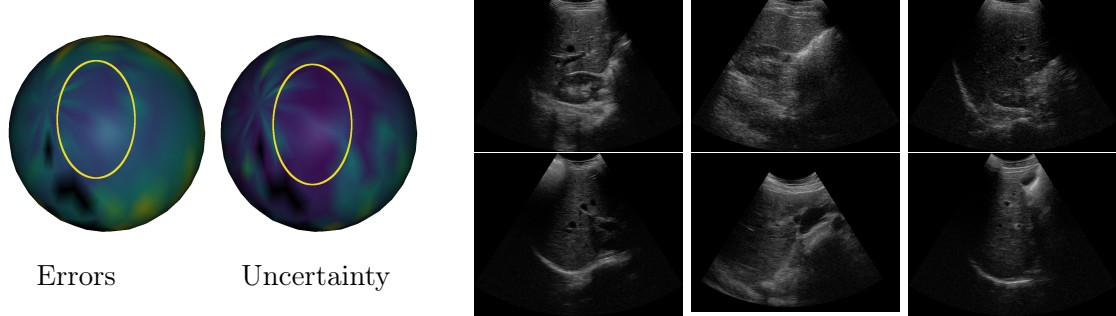

Errors       Uncertainty

Figure 7: (Left) Spheres visualizing errors and uncertainties w.r.t. spatial orientation. Colors indicate the value for the average error and uncertainty inside this region (0 for purple, up until 1.5 for yellow). For the purpose of visualization, all errors were capped at 1.5. Yellow circle shows an area with both low uncertainty and errors. (Right) Subset of images with common orientations from the circled region.

