# OpenReview forum: "Orientation Estimation of Abdominal Ultrasound Images with Multi-Hypotheses Networks"
_MIDL.io/2022/Conference — MIDL 2022_

### Official Review · Reviewer_wQtC · 2022-01-13

**Confidence:** 4
**Preliminary Rating:** 3
**Recommendation:** Poster

**Summary:**

The authors propose adapting the multi-hypothesis model by Manhardt et al., 2019, to predict the orientation of ultrasound frames relative to a target volume, and to, importantly, provide a level of confidence on those predictions. Then the most confident predictions are used to initialise the registration with a pre-operative volume. Last,  automatic segmentations of the 3D reconstructed ultrasound and the pre-op volume are used to register both ultrasound and pre-op data. The proposed method is tested on synthetic, volunteer and patient data, showing correlation between the orientation error and the estimated uncertainty.

**Strengths:**

* A nice application of multi-hypothesis network to a much needed area such as uncertainty estimation in medical imaging, and a very challenging application such as multi-modality registration

* A comprehensive evaluation using synthetic, volunteer and patient datasets.

**Weaknesses:**

* The absolute angular errors seem very large -failing to convince that the performance is good enough for a real application

* the method predicts the orientation of a frame with respect to a pre-operative volume, but only takes ultrasound data as input, hence assuming the pre-op data is in a canonical orientation with respect to the tracking data; this will require an initial calibration. This is not discussed in the paper.

* It is strange that authors decide to predict orientation and not translation; why that choice was made is unknown. It feels like translation would be easier to predict, and would equally benefit from uncertainty measurements.

**Deanonymize Review:**

yes

**Detailed Comments:**


1. p2; "Shed some light on multi-hypothesis networks," this is a very ambiguous phrasing: it is difficult to understand what kind of contribution it is to "shed some light into" something. COntributions should be concise and specific; and can be linked with conclusions, so please rephrase this contribution.

2. authors claim that the value of epsilon does not bear great importance, however they use episilon=0.6. If there is really no great impact, I would have expected to use a more "round" value, e.g. epsilon=1.

3. Angular errors are at times expressed in degrees, and at time in radians. I think they should always be expressed in the same units, and I personally prefer degrees which I find easier to make a mental picture of (especially in table 1).

4. As indicated in the discussion, absolute errors are lower in the volunteer/ clinical cases. I agree it might be due to less varied angles in the training and test sets too. For this reason, it would be most useful to report the actual ground truth angle range and distribution in the data. For example, in a linear sweep the angle might vary just a +-5 degrees during the sweep, then having  an average error of 5 degrees is actually huge; however if data was collected from all points on a sphere around an object, an average error of 5 deg is practically negligeable.

5. "In plot (a), we distribute all frames of Dataset B into different bins based on their uncertainty estimate and we compute the average error of each bin." This is an unusual way of plotting two variables to assess correlation between them. I would expect a scatter plot (without binning) and a regression analysis, or BA plot, instead. Every frame would be a point, and one axis would be uncertainty and the other would be error. The binning will distort the actual correlation, having some sort of averaging effect I guess. PLease explain why this plot was chosen and / or replace the plot to a more common one.

6. In figure 1(b) it would be most informative to add, for the 3 exemplar frames, what the actual ground truth angle is, to relate it to the error. Also I would suggest using degrees here. From Fig 1(a), it seems that the average angular error is 0.5 rad at its minimum, which is almost 30 degrees -seems a lot for the smallest error. Then, it quickly raises to .75 and beyond, which means that for most of the sequence the error is > 45 deg. I would expect this to be unacceptable in any clinical context?

7. p7: "[details are redacted for anonymity reasons, and because it is not the main focus of the paper] " Actually MIDL does not use double blind, so no need to anonymise the paper. Please include the missing statements.




**Final Rating After The Rebuttal:**

4: Weak Accept

**Justification Of The Final Rating:**

Authors have addressed satisfactorily all my concerns except for the main one, which was the exceedingly large errors in angle estimation. Specifically my concern was about how the absolute errors (in the order of 20 to 25 degrees in average) related to the arbitrary orientation of the images before reorientation. Basically, the capture range.

After querying on this during rebuttal, authors reported the error relative to the angle range ("What we can provide are the errors w.r.t. to the max spread of the dataset"). The results were showing a relative error (computed as indicated above) of > 25%. This way of comuting the relative error is, however, inadequate since it yields an overoptimistic measure. The relative error per sample should be computed relative to the ground truth value for that sample (and not the maximum of the dataset). I think this is common and simple to compute, and the only reason I can think of authors would not provide this number is because it would invalidate their results by being too large.

But precisely because what I am asking for is so simple, maybe I am missing something trivial that makes this computation impossible. Also, it seems that other reviewers did not quite pick on this as I did. I therefore give the benefit of the doubt and maintain my recommendation.


**Paper Type:**

both

**Questions To Address In The Rebuttal:**

1. Please use same units for angles throughout the paper, preferably degrees

2. Put errors in the context of the angle range in the ground truth measurements.

3. Why did the authors not predict translation also?

4. In figure 1, replace the histogram by a  scatter plot.

**Special Issue:**

no

---

### Official Review · Reviewer_FHZT · 2022-01-26

**Confidence:** 5
**Preliminary Rating:** 4
**Recommendation:** Poster

**Summary:**

The paper proposes a neural network having multiple heads trained using a meta loss that enables generating multiple predictions (multiple hypotheses of the model) to estimate the absolute orientation of abdominal ultrasound frames. Taking the standard deviation of the different hypotheses as a measure of uncertainty, the uncertainty is used to select the predictions in more confident frames for estimating the initial alignment of 3D volume (2D frames + time). In addition to good results in the evaluation with synthetic datasets of binary volumes and US images acquired with active markers used for ground truth, the proposed method provides promising results when used as initialization in the challenging task of multi-modal registration of US images and pre-op 3D MRI volume.

**Strengths:**

The paper is well written and easy to understand.

The method of using multiple heads with the meta-loss to provide a measure of uncertainty for choosing the right frames during the initialization of the registration task looks relatively simple but effective.

The experiments are done in three different setups of varying difficulty but show that the proposed method performs consistently well.

**Weaknesses:**

The results section shows that there is a large difference with MC dropout, but it is not clear why this should be. Any potential reasons one could speculate or any intuition? A baseline comparison against methods that use different representation than the quaternion for doing statistics on the Lie group SE(3), or estimating orientation using landmarks for alignment would be helpful to see if MC dropout is better adapted in these setups compared to the ones proposed here.

**Deanonymize Review:**

no

**Detailed Comments:**

MIDL submission is single-blind and since reviewers can see the authors' names, not sure why there was a need for anonymity in redacting the details of the registration experiments.

Minor typos:

Abstract: “..., but also to produce …" -> using and instead of but also; or perhaps add, “In this work, we not only train …" if using but.

**Final Rating After The Rebuttal:**

4: Weak Accept

**Justification Of The Final Rating:**

The authors have responded and addressed my queries, and in my opinion, reasonably addressed other reviewers' comments as well. The paper will likely be of interest to the MIDL community, and hence I am sticking with my rating.

**Paper Type:**

both

**Questions To Address In The Rebuttal:**

I would like see more discussion, even if it is a speculation or an educated guess, on why MC dropout should provide worse results than the proposed method.

The redacted details of the multi-modal experiments.

**Special Issue:**

no

---

### Official Review · Reviewer_ywFa · 2022-01-27

**Confidence:** 4
**Preliminary Rating:** 4
**Recommendation:** Poster

**Summary:**

The authors propose a method to predict the orientation of 2D abdominal ultrasound images and simultaneously output the confidence of the prediction. A neural network is trained to produce multiple hypotheses (i.e. orientation estimation) and a meta-loss is used to optimise the network's weights. The confidence of the prediction can then be derived during inference by computing the standard deviation (or other statistical metrics) between the multiple hypotheses. The proposed method is compared with Monte-Carlo dropout and network ensembling on 3 datasets. The authors show that the proposed framework yields comparable or superior uncertainty estimation (i.e. correlation between the uncertainty and prediction error), with the added advantage of being easier to implement. The authors also demonstrate its application to multi-modal image registration.

**Strengths:**

-	The proposed approach is simple, intuitive yet effective for confidence estimation. The performance is comparable to the existing methods while being easier and faster to implement, which can be a good baseline for future studies
-	The experiments are thorough. The proposed approach is compared with two of the most popular approaches on three different datasets. The orientation prediction accuracy, confidence and training and testing time are all well reported. Its application on initializing registration is also demonstrated and tested.
-	The limitation of generalizability is addressed clearly, which further shows the extra potential application of the proposed method to out-of-distribution detection.
-	The whole paper is clearly written, with an intuitive explanation of the meta-loss and a detailed appendix which provides a lot of useful supplementary information for better understanding of the experimental setup and results.


**Weaknesses:**

-	The technical novelty is limited as the meta-loss and multiple hypotheses used in the paper were already proposed in prior work. But this is a minor point because, as pointed out by the authors, the application of these techniques on ultrasound orientation estimation is novel and appropriate. Nevertheless, the technical difference, if any, should be analyzed and discussed in more details.
-	Section 2.3, especially the Rotation part, is not clearly written. In the same section, the authors mention that two extra segmentation networks are needed for the registration, which seems to be a significant limitation for the application (e.g. in terms of data and resources needed for training) but the authors provide very little information on that.
-	There is no figure illustrating the proposed pipeline/method. This was possibly due to space constraints, but I believe it would improve clarity.


**Deanonymize Review:**

no

**Detailed Comments:**

-	Certain omissions from the main text are actually covered in the Appendix, which may cause readers to miss important information if they miss the Appendix because the two parts (i.e. Main Text and Appendix) are not explicitly linked. Examples are choices of hyperparameter (M and epsilon), acquisition and examples of each dataset, as well as the correlation between the uncertainty and anatomical structures presented in the image.
-	The discussion in Section 4.1.1 could make better reference to the results. When referencing the results presented in Table 1, the authors should point out clearly which parts of the table they refer to.


**Final Rating After The Rebuttal:**

4: Weak Accept

**Justification Of The Final Rating:**

I wish to thank the authors for adequately addressing my comments in their rebuttal, and those of the other reviewers. In my opinion, this work was well conducted and should be accepted as it will be of interest to the MIDL community.

**Paper Type:**

both

**Questions To Address In The Rebuttal:**

-	Please see comments in the “detailed questions” section above.
-	A more detailed analysis or discussion on the technical difference between the multi-hypotheses networks and this paper, for example how the pose estimation in RGB images different from this application and hence lead to the difference in the methods proposed, if any.
-	Section 2.3 and sections corresponding to the registration can be described more clearly
-	(space permitting) Including a figure to illustrate the high-level proposed framework


**Special Issue:**

no

---

### Official Review · Reviewer_TeLX · 2022-01-28

**Confidence:** 5
**Preliminary Rating:** 2
**Recommendation:** Poster

**Summary:**

This paper addresses orientation regression of 2D ultrasound sweeps with multi-head networks for the downstream task of multi-modal image registration. The $ M $ heads are trained using a meta loss adopted from Manhardt et al. (2019). The disagreement of the multiple predictions from the different heads are interpreted as predictive confidence. The presented approach is evaluated on three data sets ranging from toy data to real clinical data and it is shown that $ M $-head networks yield low angular errors and confidence estimates that are moderately correlated with the predictive error by means of Spearman's correlation. They seem to perform similar to full network ensembles but with less computational cost. Further, it is observed that the mean $ M $-head confidence is considerably higher on out-of-distribution data. Finally, the the orientation estimate is used to initialize multi-modal registration between the inter-operative US and a pre-operative MR scan. Employing the orientation estimation lowers the final registration error on the clinical data set.

**Strengths:**

* Consideration of predictive uncertainty is of high importance in medical image analysis and still often overlooked.
* The experimental evaluation is methodologically correct.
* The presented approach is compared to other methods for uncertainty estimation.
* The paper is easy to follow.

**Weaknesses:**

* No methodological novelty is presented and the overall structure of the paper, including the comparison methods, exhibits great resemblance to Linmans et al. (2020).
* The reported errors seem quite large, potentially prohibiting a real world application.
* It is unclear to me how the confidence is actually computed. It is unsurprising that the eigenvalues/singular values yielded very similar results as the standard deviation, as the singular values of a data matrix denote the variance of the data along the axis of the corresponding singular vector. § 2.2.2 would probably benefit from more equations.
* Manhardt et al. (2019) actually used the SVD to compute the axis of ambiguity for, e.g., rotationally symmetric objects. It seems that the authors blindly followed that without proper interpretation of what the computed confidence stands for.
* The use of correlation to assess the quality of uncertainty goes in the right direction. However, an in-depth analysis of the uncertainty calibration should have been performed. Proper metrics for regression exist, e.g., (Levi et al., 2019; Laves et al., 2021).
* The runtime comparison of MC dropout is somewhat unfair, as only the last layer with dropout has to be evaluated multiple times and not the entire network. This should result in similar runtimes as multi-hyp.
* Both the ensemble and multi-hyp networks have considerably more parameters as the network trained with MC dropout. For a fair comparison, the number of parameters should at least be adjusted for the dropout rate.
* I think that the ensembles could perform much better as they were trained using data subsets. Lakshminarayanan et al. (2017) pointed out that using different random initialization and training on the full data set performs quite well. The necessary randomness can be injected by different sample selection in SGD.
* The downstream task in § 4.2 is not sufficiently described and lacks many details or references.

Levi, D., Gispan, L., Giladi, N., & Fetaya, E. (2019). Evaluating and calibrating uncertainty prediction in regression tasks. arXiv:1905.11659.

Laves, M. H., Ihler, S., Fast, J. F., Kahrs, L. A., & Ortmaier, T. (2021). Recalibration of Aleatoric and Epistemic Regression Uncertainty in Medical Imaging. Journal of Machine Learning for Biomedical Imaging. arXiv:2104.12376.

Lakshminarayanan, B., Pritzel, A., & Blundell, C. (2017). Simple and scalable predictive uncertainty estimation using deep ensembles. NeurIPS 2017 arXiv:1612.01474.

**Deanonymize Review:**

no

**Detailed Comments:**

Some minor comments:

* The angular error should always be given in the same unit.
* It could be noted that $ M $ head networks and full ensembles are both ensemble networks sitting at the opposite ends of a shared weight spectrum.
* Eq. (2) uses $ \hat{\mathbf{q}} $ to denote the ground truth. However, most literature uses the hat to denote the *predicted* value.
* Tenses of the written text seem inconsistent (e.g., past tense in § 2.1).
* Many sentences have unsual syntax that reduces readability, e.g.:
  * "Our method produces on a clinical test estimates ..."
  * "We acquired for the purpose of this study a dataset ..."
  * "This means that they are all able to select via their confidence estimation a better frame..."

**Final Rating After The Rebuttal:**

3: Borderline

**Justification Of The Final Rating:**

The authors did a good job of answering my questions by performing additional experiments during the very limited rebuttal phase. Therefore, I increase my rating. However, for an application paper, I miss a stronger focus on the downstream task of multi-modal registration and the exploitation of the uncertainty estimates. The correlation assessment and OOD detection would be sufficient if a novel method were presented. The findings provide little novelty over what was already reported by Linmans et al. (2020). It would be very interesting to see if the uncertainty can be used to improve the actual registration.

One minor comment: In the present conclusion, the authors state that ensembles do not perform significantly better than multi-hyp nets. However, no statistical hypothesis tests appear to have been performed and no $p$-values are reported.

**Paper Type:**

validation/application paper

**Questions To Address In The Rebuttal:**

* I wonder how ensembles compare if trained with the same meta loss and how multi-hyp compares if trained on subsets.
* What do the values behind the $ \pm $ in Tab. 1 refer to and why are they missing for data set A?
* See also weaknesses.

**Special Issue:**

no

---

### Meta-Review · Area_Chair_RQiC · 2022-02-16

**Recommendation:** Accept (Poster)
**Confidence:** 5

**Metareview:**

For this paper, there was a very active rebuttal phase, where the authors appropriately addressed most of the reviewers' concerns. One reviewer would have liked to see a stronger focus on downstream tasks. However, there is general consensus that the work was conducted well and merits publication, and I agree and recommend acceptance.

---

### Decision · Program_Chairs · 2022-02-28

Accept